# Understanding Necroptosis in Pancreatic Diseases

**DOI:** 10.3390/biom12060828

**Published:** 2022-06-13

**Authors:** Ru He, Zhengfeng Wang, Shi Dong, Zhou Chen, Wence Zhou

**Affiliations:** 1The First School of Clinical Medical, Lanzhou University, Lanzhou 730000, China; her20@lzu.edu.cn (R.H.); wangzf20@lzu.edu.cn (Z.W.); dongsh20@lzu.edu.cn (S.D.); chenzh20@lzu.edu.cn (Z.C.); 2Department of General Surgery, The First Hospital of Lanzhou University, Lanzhou 730000, China; 3Department of General Surgery, Lanzhou University Second Hospital, Lanzhou 730000, China

**Keywords:** necroptosis, pancreatitis, pancreatic cancer, RIPK, MLKL, TNF

## Abstract

Intermediate between apoptosis and necrosis, necroptosis is a regulated caspase-independent programmed cell death that induces an inflammatory response and mediates cancer development. As our understanding improves, its role in the physiopathology of numerous diseases, including pancreatic diseases, has been reconsidered, and especially in pancreatitis and pancreatic cancer. However, the exact pathogenesis remains elusive, even though some studies have been conducted on these diseases. Its unique mechanisms of action in diseases are expected to bring prospects for the treatment of pancreatic diseases. Therefore, it is imperative to further explore its molecular mechanism in pancreatic diseases in order to identify novel therapeutic options. This article introduces recent related research on necroptosis and pancreatic diseases, explores necroptosis-related molecular pathways, and provides a theoretical foundation for new therapeutic targets for pancreatic diseases.

## 1. Introduction

Necroptosis is a form of cell death that maintains intracellular homeostasis when the apoptotic pathway is altered or suppressed [1]. It may be caused by chemical and physical injury, inflammation, or infection, resulting in the rupture of the cell membrane, gradual translucent cytoplasm, and the swelling of organelles, thereby leading to the release of cell contents, the exposure of damage-associated molecular patterns, as well as the triggering of an inflammatory reaction [2]. Additionally, necroptosis plays a key role in the biological regulation of cancer, including tumor occurrence, metastasis, and cancer immunity [3]. As a combination of apoptosis and necrosis, it plays a dual role in cancer progression, promoting tumor proliferation, invasion, and metastasis, and it can also play the opposite biological function [4]. A considerable number of studies have established that targeting this pathway can modulate the progression of diseases (including promotion or inhibition), such as nervous inflammation in the nervous system [5], viral inflammation [6], myocardial infarction [7], liver diseases (acute liver injury, chronic hepatitis, cirrhosis, and hepatocellular carcinoma) [8], cholangiocarcinoma [9], and other diseases. Furthermore, necroptosis also inhibits tumor re-deterioration following radiotherapy. For example, the activation of the receptor-interacting serine/threonine-protein kinase 1 (RIPK1)/RIPK3/mixed lineage kinase ligand (MLKL)/C-Jun N-terminal Kinase (JNK)/IL-8 pathway can block tumor-cell reaggregation in colorectal cancer after radiotherapy [9]. On the basis of the above results, the necroptotic pathway may become a new therapeutic strategy for diseases. 

The aberrant exocrine function of pancreatic glands typically leads to acute pancreatitis (AP), chronic pancreatitis (CP), and pancreatic cancer (PC). Glandular damage from repeated episodes of AP can result in irreversible changes, eventually leading to CP. A huge body of evidence suggests that CP is a risk factor for PC [10,11]. AP and CP and PC impose not only psychological and financial burdens on patients, but also huge health and economic burdens on society. However, the new mechanisms to identify these diseases have been slow [12]. The latest studies report on the chief role of necroptosis in pancreatitis and PC [13,14]. Therefore, in this review, we focused on the necroptotic pathway between pancreatitis and PC, summarized the molecular mechanism of necroptosis and its influence on the development of pancreatitis and PC, and explored the effect of related molecular compounds in the treatment of diseases in order to provide a foundation for the development of effective treatment methods for pancreatic diseases in the later stage.

## 2. Necroptosis

### 2.1. Necroptosis Signal Transduction

Necroptosis and apoptosis share a similar mechanism and cell morphology [15]. Intracellular and extracellular stimuli and corresponding ligands of the death receptor family trigger necroptosis [16]. Tumor necrosis factor α (TNF-α), tumor necrosis factor receptor superfamily members, fas ligand complex (Fasl), viral DNA/RNA, lipopolysaccharide (LPS) and interferon (IFN) can bind to toll-like receptor (TLR), tumor necrosis factor receptor 1 (TNFR1), fas, Z-DNA binding protein 1 (ZBP1), and other receptors [17,18,19], altering downstream signaling pathways and leading to necroptosis. The proteins implicated in signal transduction include RIPK1, RIPK3, and MLKL, which form different protein complexes that participate in the necroptotic pathway [20]. Moreover, small molecular compounds, such as necrostatin-1 (Nec-1), Nec-4, and TAK-632, are thought to repress the necroptotic pathway [21]. Currently, the detection of phosphorylated RIPK1, RIPK3, or MLKL, as well as the oligomerization and membrane localization of MLKL, are considered biomarkers of necroptosis in vitro [22].

#### 2.1.1. TNF-α/TNFR Pathway

Among the stimuli described above, the TNF-α/TNFR stimulation pathway is the most studied [23]. Following TNF-α stimulation, TNFR1 translocates to the lipid membrane and recruits RIPK1, TNFR-associated death domain (TRADD), a cellular inhibitor of apoptosis protein 1 (cIAP1) and cIAP2, TNFR-associated factor 2 (TRAF2), and TRAF5 to form Complex I [24]. After the formation of Complex I, under the influence of different signaling complexes, the next cell survival or death can be determined [25]. Under the action of deubiquitinating enzymes (DUBs), RIPK1 restricts the activation of the Nuclear factor kappa-B (NF-κB) pathway and forms Complex II, composed of RIPK1, TRADD, caspase-8 (Casp-8), and FAS-associated death domain (FADD) [26]. On the one hand, when caspase-8 is activated, a caspase cascade reaction is subsequently initiated, leading to the apoptotic pathway [27]. On the other hand, when a caspase-8 inhibitor is used, the necroptotic pathway is activated. Thereafter, the autophosphorylation of RIPK1 is induced, and RIPK3 is phosphorylated via its RIP homotypic interaction motifs (RHIMs) in order to form key necrosis bodies [28]. In necrosomes, the PsKD phosphorylation of MLKL in the kinase-like domain activates the ring of RIPK3, leading to its oligomerization [29]. This conformational conversion translocates MLKL to the plasma membrane, and intracellular osmotic pressure is increased by triggering the influx of Na^+^ and Ca^2+^. Afterward, the membrane pore opens, releasing damage-associated molecular patterns (DAMPs) and bringing about tissue inflammation and organ damage [30,31]. The mechanism underlying the mediation of necroptosis by mitochondrial reactive oxygen species (ROS) involves the activation of RIPK1 autophosphorylation on serine residue 161 (S161) by ROS and the recruitment of RIPK3 to form necrosomes [32]. Fas, another death receptor, also stimulates necroptosis through the activation of RIPK3 and downstream MLKL in the presence of caspase inhibitors [33].

#### 2.1.2. Toll-like Receptor Pathway

Unlike necroptosis induced by the TNF-α/TNFR pathway, the TLR responds to the stimulation of multiple pathogens. TLR3 can detect viral double-stranded RNA (dsRNA), and TLR4 can be stimulated by LPS [34,35]. In a study assessing the relationship between LPS-induced intestinal injury and necroptosis, the visualization of the jejunum using electron microscopy signaled that the LPS-induced intestinal injury was characterized by typical necroptosis. Moreover, the expression levels of RIPK1, RIPK3, and MLKL proteins were increased. In the jejunum pretreated with Nec-1, the necroptosis-induced morphological damage was diminished, while the intestinal digestion and barrier functions were enhanced [36]. Multiple studies have reported that LPS stimulated TLR3/TLR4 or detected viral DNA through the DNA-dependent activators of Z-DNA binding protein 1 (ZBP1), resulting in the formation of a toll-like receptor domain adaptor protein (TRIF)–RIPK3 complex, ZBP1–RIPK3 complex, or RIPK1 nonbinding necrotic body. Phosphorylated MLKL was transported to the plasma membrane, resulting in membrane rupture, DAMPs release, and necroptosis [37]. 

#### 2.1.3. ZBP1-Mediated Pathway

Under viral stimulation, Z-RNA/DNA generated by viral replication activates ZBP1 in the nucleus, inducing RIPK3 in the nucleus to mediate MLKL activation, resulting in nuclear-membrane rupture, RNA leakage into the cytoplasmic matrix, and eventually leading to necroptosis [38,39]. Similarly, Yang et al. [19] described that IFN-induced necroptosis was also mediated by ZBP1–RIPK3 and was associated with systemic inflammatory response syndrome. Studies have also validated that IFN activates the RIPK1–RIPK3 necrosome complex via Jak/STAT signaling transduction to induce necroptosis when FADD is absent or not phosphorylated, or when caspases are inactivated [40]. In ovarian cancer, Jak/STAT 3 is a promising therapeutic target, as it is frequently overexpressed. Jak/STAT 3 plays a role in tumor angiogenesis, survival, invasion, and chemoresistance. The inhibition of this pathway is shown to suppress tumor growth and progression [41]. The following figure illustrates the necroptosis-related signaling pathways (Figure 1).

### 2.2. Other Forms of Cell Death

Cell death contributes to cellular and tissue homeostasis and plays a central role in various diseases. A growing number of researchers are currently focusing on studying cell-death methods for disease treatment. In addition to the necroptotic process described above, major forms of cell death include apoptosis, pyroptosis, ferroptosis, autophagy, parthanatos, entotic cell death, NETotic cell death, immunogenic cell death, and copper-dependent death and anoikis [42,43,44,45,46]. Apoptosis, also known as programmed cell death, plays a crucial role in organ development and homeostasis and is often associated with developmental disorders and tumorigenesis [47]. Apoptotic cells are characterized by plasma-membrane infiltration, mitochondrial-outer-membrane permeability, DNA fragmentation, nuclear disintegration, and finally, cellular disintegration into apoptotic bodies, which are phagocytosed by immune cells and surrounding cells [48]. At present, the ability of malignant cells to escape apoptosis is considered one of the hallmarks of cancer [49]. Pyroptosis is an inflammatory cell death mediated by caspase-1 and is usually caused by microbial infection. It is defined by cellular swelling and plasma-membrane rupture, leading to the release of the proinflammatory cytokines IL-1β and IL-18, and cell content into extracellular space, thereby inducing an inflammatory response [50]. Furthermore, it is related to innate immune diseases and regulates the proliferation and migration of cancer regulated via noncoding RNA and other molecules, which has major implications for the prevention and treatment of cancer [51]. The identification of microRNAs (miRNAs) and long noncoding RNAs can provide valuable information in diagnosis and prognosis and offers predictive value in several malignancies, such as melanoma. Their increase can be related to either the cancer or other conditions, such as inflammation. The therapeutic targeting of miRNAs can impact the natural history of melanoma by enhancing the sensitivity to both standard therapies and immune checkpoint inhibitors [52]. The dense extracellular matrix (ECM) in PC tissue is widely considered to be a natural barrier, limiting antiangiogenic therapy, as well as chemotherapy and immunotherapy [53]. In contrast, anoikis is a specific type of apoptosis-like form of cell death in response to inappropriate cell–ECM interactions, and it plays a key role in the prevention of metastatic colonization during tumor invasion and metastasis [54]. Other forms of cell death have not been discussed in detail herein; you may wish to refer to other pieces of literature on the subject (Table 1).

## 3. Pancreatitis

Pancreatitis refers to the indigestion of the pancreas itself, in which pancreatic enzymes damage pancreatic tissues and cause glandular, organ, and system dysfunction. On the basis of the duration of the disease, pancreatitis can be classified into AP and CP. According to the severity of the disease, AP can further be divided into mild, moderate, and severe. Repeated episodes of AP are referred to as recurrent acute pancreatitis, which eventually develops into CP driven by chronic alcohol consumption or genetic risk factors [65,66]. Patients with pancreatitis often deteriorate and develop necrotizing pancreatitis, and the underlying causes still remain to be elucidated. However, the transformation of cell-death patterns, such as apoptosis and necrosis into necroptosis, may explain this phenomenon.

### 3.1. Pathophysiology of Pancreatitis

Owing to the limitations of the existing experimental conditions, isolated acinar cells and animal models cannot be preserved for a long time. As a result, the majority of experiments mainly focus on AP, and there are insufficient experimental data to confirm the relationship between the mechanisms described below and CP. Nevertheless, some researchers speculate that these mechanisms are linked to CP, with the expectation that these hypotheses will be verified in CP once the experimental equipment and technology of the future have matured.

#### 3.1.1. Trypsin Activation

The activation of trypsin is considered one of the earliest events in the occurrence of AP. Early intracellular trypsinogen activation may lead to the development of AP or its transformation into CP [67,68]. Ca^2+^ is known to play a primary role in the activation of intracellular trypsin. Irrespective of the etiology, all cases of APs are characterized by a steady increase in intracellular Ca^2+^ [69]. On the one hand, under the stimulation of bile, alcohol, intraductal hypertension, and other factors, IP3Rs (inositol trisphosphate) and RyRs trigger the release of a large amount of Ca^2+^ stored in the endoplasmic reticulum, which subsequently penetrates the acinar cells [70,71]. This results in the instability of secretory enzyme granules, intracellular vacuolization [72], and calcium-dependent trypsin activation. Finally, mitochondrial damage is induced, cell apoptosis and necrosis are activated, and acinar cells become necrotic, resulting in inflammatory reactions [73]. On the other hand, if the release of pathological calcium cannot be controlled, rapid energy consumption occurs, and the cells become necrotic. Cell components leak out of the cells, and white blood cells identify and activate the inflammatory corpuscle signaling pathway; IL-1β and TNFα are released [74]. When TNFα reaches the basal lateral membrane of unaffected or marginally damaged acinar cells, it induces necroptosis. Interestingly, intracellular Ca^2+^ overload also occurs in necroptosis. The connection between the two has not yet been reported in the relevant literature, and thus, additional evidence is necessitated to elucidate their relationship.

#### 3.1.2. Systemic Inflammatory Response

##### NF-κB

NF-κB is a transcriptional factor that regulates inflammatory responses [75]. In pancreatic acinar cells, the NF-κB signaling pathway is also induced by intracellular Ca^2+^ inflow [76], together with the phosphorylation of IκBα. Next, the proteasome degradation and nuclear translocation of NF-κB (p65/p50) trigger the release of cellular IL-6 and TNFα, thereby resulting in inflammation [77].

Recent studies have demonstrated that the continuous activation of the NF-κB pathway in acinar cells seems to be a key pathogenic mechanism underlying chronic pancreatitis and is unrelated to trypsinogen activation. However, a study revealed that the knockout of NF-κB p56 (a component of NF-κB) in pancreatic acinar cells promoted fibrosis in a CP mouse model induced by cerulin. Meanwhile, p65 knockout in macrophages lowered the production of cytokines and alleviated fibrosis in CP [78]. In contrast, Huang et al. [79] found that the continuous activation of NF-κB in pancreatic acinar cells exacerbated the severity of CP. Other researchers corroborated that NF-κB small-molecule inhibitors prevented endoplasmic-reticulum stress and NOD-like receptor thermal protein domain associated protein 3 (NLRP3) inflammatory cytokine activation, and delayed pancreatitis progression [80]. In addition to NF-κB activation, other types of acinar-cell damage may have contributed to the discrepancies in the abovementioned findings, and gene knockout may have played a role.

##### Infiltrating Immune Cells

In pancreatitis, cytokines predominantly secreted by acinar cells (TNFα, IL-6) or DAMPs mediate the activation of immune cells (such as neutrophils and macrophages), thereby increasing pancreatic damage and ultimately leading to systemic inflammation [81,82]. Associated with necroptosis, acinar cells secrete TNFα, and necroptosis is induced by TNFα stimulation. The rupture of necroptotic cell membranes also releases large amounts of DAMPs and cytokines to promote inflammatory progression [83]. On the basis of the above results, a positive feedback loop may be formed during the progression of pancreatitis and may further accelerate its progression. Wang et al. [84] outlined the different mechanisms of necroptosis and summarized its significance in various tissue inflammations and diseases. Even though necroptosis has been extensively explored, studies on necroptosis associated with immune cells are scarce. Given its important role in diseases, it is worth further exploring the mechanism and triggering factors of immune-cell necroptosis in order to gain a better understanding of this pathway and play a greater role in future treatments.

### 3.2. Necroptosis and Pancreatitis

AP typically manifests as persistent pain in the upper abdomen. The diagnosis is mainly based on: (1) clinical manifestations: abdominal pain consistent with AP (persistent, upper abdominal pain, and referred pain to the back); (2) serum amylase or lipase at least three times higher than the normal value; and (3) imaging examination (CT, MRI, or abdominal ultrasound) [66,85]. Among all types of pancreatitis, AP is the most commonly encountered. The incidence of AP is estimated to be 34/100,000 (95% CI 23–49), with a mortality rate of 1.16/100,000 (95% CI: 0.85–1.58) [86]; the disease is characterized by local or systemic inflammatory responses. Most patients exhibit self-limiting mild AP that usually resolves within a week. Approximately 20% of patients develop moderate or severe acute pancreatitis (SAP), with necrosis of the pancreas, surrounding tissue, or organ failure, and the mortality rate is as high as 20–40% [86,87,88].

Considering the clinical manifestation of AP and the high mortality rates associated with comorbidities, timely diagnosis and treatment are critical. Studies have exposed that the degree of necrosis in acinar cells determines the severity of pancreatitis [89]. Necroptosis, as a newly discovered cell-death mode, has gradually attracted attention in inflammatory diseases. Studies have demonstrated that the GNE684 (an effective inhibitor of RIPK1) slows the progression of some inflammatory diseases [90]. It has been reported that, in AP, the expression levels of RIPK3 and phosphorylated MLKL were positively correlated with the degree of necrosis, whereas that of RIPK1 was negatively correlated with the degree of necrosis. The knockdown of RIPK1 inhibits NF-κB activation and promotes acinar-cell necrosis. Consequently, the overexpression of RIPK1 suppresses the inhibition of RIPK3, decreases the phosphorylation level of MLKL, and reduces the necrosis of acinar cells, thereby playing a protective role in AP mice [91]. However, in SAP mice, the expression levels of RIPK1, RIPK3, and p-MLKL in the experimental group were substantially higher than those in the control group [92]. This is most likely due to the increased degree of pancreatic necrosis in SAP. Louhimo et al. [93] constructed an SAP mouse model and determined that the main form of pancreatic-acinar-cell death was necroptosis, and the severity of pancreatitis could be minimized via the inhibition of necroptosis through the loss of somatostatin or RIPK3. Furthermore, the administration of necrostatin following pancreatitis can alleviate cellular injury. This implies that the inhibition of necroptosis may be an effective therapy for severe pancreatitis. However, in another caerulein-induced-pancreatitis study using RIPK3- or MLKL-deficient mice, compared with the control group, the pancreatic edema and inflammation in RIPK3−/− and MLKL−/− mice were more severe, and more inflammatory cells were recruited into the pancreas. Moreover, the MLKL deletion resulted in the downregulation of antiapoptotic genes and thus increased the apoptotic rate [94]. The above findings indicate that RIPK3- or MLKL-mediated necroptosis plays a protective role in AP. Previous studies on pancreatitis and necroptosis have reported conflicting outcomes, which may be attributed to the induction medication, treatment duration, and observation time window for necroptosis. Nonetheless, necroptosis is unquestionably closely related to pancreatitis progression, and additional research is required to authenticate their relationship.

#### Treatment of Pancreatitis Based on Necroptosis

RIPK1 is a key component of necroptosis and determining the role of its inhibitor, Nec-1, in acute pancreatitis is essential. Multiple studies have unraveled that Nec-1 is effective at inducing AP. However, RIPK1 kinase modification (RIPK1K45A: kinase-dead) only functions in some mice models, indicating that RIPK1 may play a secondary role in mediating necroptosis [95]. Later on, other studies also exposed that, when activated RIPK1 in acinar cells is inhibited, AP mice might mediate RIPK1-dependent necrotizing regulation through the RIPK1/NF-κB/AQP8 axis, thereby regulating the changes in the degree of local and systemic inflammation [96]. Considering the significant increase in RIPK3 under pathological pancreatitis conditions, and the correlation between pancreatitis and necroptosis, the same formulation of non-HA-binding liposomes containing shRNA mRIPK3-pDNA effectively controlled disease progression by reducing the serum amylase concentration and inflammatory-cell infiltration [97]. Serum amyloid A (SAA) is an early sensitive biomarker of inflammatory diseases. Yang et al. [98] showed that SAA3 induces the RIPK3-dependent necroptotic pathway in acinar cells and causes AP, suggesting that the pathway might be a potential drug target for AP.

Growing evidence suggests that miRNAs can improve the prognosis of pancreatitis in experimental mouse models. miR-21 is the most studied miRNA, and its high expression was detected in AP mouse models. Earlier studies have confirmed that its inhibition in wild-type mice alleviated the severity of AP and protected against tissue damage and necroptosis induced by TNF-α, which supports the fact that miR-21 is a miRNA that promotes necroptosis [99]. Jia et al. [100] first verified that miR-325-3p was downregulated in AP patients and mouse models and decreased with an increase in the disease severity. It was then demonstrated that miR-325-3p inhibited the RIPK3/MLKL signaling pathway in acinar cells to slow down the development of AP. The above two studies indicate that miR-21 and miR-325-3p are related to the necroptosis-dependent pathway in pancreatitis and, therefore, can be used as potential therapeutic targets to develop drugs that prevent necroptosis. Bone marrow mesenchymal stem cell (BMSC) treatment reduced the TNF-α-induced necroptosis of pancreatic acinar cells in vitro and significantly decreased the severity of SAP in mice. Indeed, BMSCs can reduce system inflammatory responses by inhibiting cytokines, such as IL-1β, IL-6, and TNF-α; increasing the levels of anti-inflammatory mediators, such as IL-4 and IL-10; and regulating the necroptotic pathway mediated by RIPK1/RIPK3/MLKL [92]. By combining BMSC and miRNA, overexpressed miR-9 (miR-9-BMSCs) significantly reduced systemic inflammatory responses, hindered the necroptotic signaling pathway, and promoted the regeneration of the injured pancreas in vivo. miR-9-BMSCs secrete miR-9, targeting RIPK1 in pancreatic acinar cells induced by TNF-α and inhibiting necroptosis by improving the SAP [101].

With adjustments in dietary habits, pancreatitis caused by hyperlipidemia is becoming more and more common. Hong et al. [102] found that a high-fat diet aggravated the scope and severity of AP in mice. However, the use of the TLR4 selective inhibitor TAK-242 inhibited oxidative stress, alleviated the inflammatory response and necroptosis, and delayed the progression of AP in mice. CaMK II is a calcium-modulating protein that is upregulated in the pancreatic acinar cells of AP mice. It has been discovered that its inhibitor, KN93, protected against AP in mice by reducing oxidative stress products, reducing the expression levels of the RIPK3 and p-MLKL pathways and preventing necroptosis [103]. Some studies have evinced that inhibiting Hif-1α also alleviates AP by reducing the production of ROS and regulating the necroptosis signaling pathway [104]. These results suggest that inhibiting oxidative stress may also be a promising therapy for AP in clinics.

Fluid resuscitation is the first-line treatment for shock in SAP. However, intestinal ischemia–reperfusion injury occurs after fluid resuscitation, causing systemic inflammatory response syndrome and multiple organ dysfunction syndrome. Cui et al. [105] demonstrated that the administration of the necroptotic inhibitor Nec-1 prior to active fluid resuscitation can reduce the levels of HMGB1 (determining intestinal barrier dysfunction and infection in SAP patients) in serum and cytoplasm and can thus alleviate the development of SAP-related organs in mice.

### 3.3. Necroptosis and Chronic Pancreatitis

CP is an irreversible pancreatic tissue injury that is provoked by recurrent pancreatitis of different intensities. Alcohol and tobacco are the most common risk factors for CP [106]. However, data on the epidemiology and etiology of CP are limited, heavily dependent on regional standards, and prone to reporting errors [107]. Furthermore, there is no gold standard for the diagnosis of CP. The most useful diagnostic approaches may be detailed history taking and thorough physical examination, while CT or MRI is currently recommended as the first-line test for CP. In contrast, endoscopic ultrasonography (EUS) is recommended for problematic diagnoses after cross-sectional imaging, and contrast-enhanced magnetic resonance cholangiopancreatography (S-MRCP) can also be used if the diagnosis is not clear. Finally, histological examination can be used as the gold standard for the diagnosis of CP in high-risk patients [108]. With the increase in CP incidence and prevalence, there is no effective treatment. The expression levels of LPS in the pancreatic tissues of mice, healthy patients, and the control group were determined by immunohistochemistry (IHC) and Western blotting (WB), and necroptosis and trypsin activation were amplified in mice after LPS injection, which supported the potential role of LPS in the pathogenesis of CP [109]. LPS can also induce the necroptosis pathway and exacerbate the inflammatory response. This represents the potential value of necroptosis in CP progression and treatment. However, the relationship between the two has so far not been determined. There are other reports about chronic inflammation and necroptosis, such as chronic hepatitis [7] and neuroinflammation [110].

## 4. PC

### 4.1. Necroptosis and Pancreatic Cancer

As is well documented, PC is a fatal [111], highly aggressive, and chemoresistant disease. With a surge in its risk factors, such as aging, smoking, obesity, diabetes, and alcohol intake, the incidence and morbidity of PC have increased and have gradually become the leading cause of cancer-related deaths worldwide [112]. According to the Global Cancer Observatory (GLOBOCAN) 2020, it is estimated that a total of 495,773 patients worldwide will be diagnosed with PC in 2020, while 466,003 may die from PC [113]. Currently, the prognosis of PC remains poor. The 5-year survival rate is roughly 5–10%, and life expectancy is less than 5 months after diagnosis [114]. Serum carcinoembryonic antigen (CA19-9) is the only biomarker approved for the routine detection of PC [115]. However, due to its low positive rate in asymptomatic patients, it is only used to monitor the response to treatment or as a marker for recurrent diseases [116]. As a malignant tumor with a poor prognosis and no effective biomarkers, it is urgent to reveal the mechanism by which PC progresses, formulate effective treatment strategies, and improve the overall survival rate of patients [117]. Preclinical studies have reported that resistance in colorectal cancer caused by caspase-8 deficiency can be overcome by inducing necroptosis [118]. However, RIPK1 and RIPK3, the major components of necrosomes in PC, were upregulated, and cell immunity was induced by C-X-C motif chemokine ligand 1 (CXCL1) and Mincle to promote tumor growth [119]. Ando et al. [120] also showed that the expression levels of RIPK3 and MLKL in cancerous tissues were higher than those in normal pancreatic tissues using IHC and WB analysis. A conditioned medium derived from necroptotic cells promoted cancer-cell migration and invasion through the CXCL5–CXCR2 axis. However, multiple studies have exposed that the inhibition of RIPK1 had no effect on tumor growth or survival in Kras-driven pancreatic tumor models, and it did not reduce lung metastasis in a B16 melanoma model [90]. As Xia et al., describe in the article, necroptosis plays a double-edged-sword role due to its antitumor and tumor-progression properties [4]. Overall, the role of necroptosis varied across studies. The specific role of necroptotic drugs in PC, and their potential as a therapeutic strategy for PC, warrant further investigations.

### 4.2. Treatment of Pancreatic Cancer Based on Necroptosis

Necroptosis can occur under various types of stimulation. On the basis of its signaling transduction pathway, necroptosis can be modulated in multiple ways to slow down tumor progression. TRAIL-stimulated necroptosis induces necroptosis in PC cells under ROS inhibition, reduces inflammation in cancer, and inhibits tumor progression [121]. The expression level of the aurora kinase AURKA was observed to be increased in patients with PC and was associated with poor overall survival. Its inhibitor, CCT137690, induced the necroptosis of PC cells via RIPK1, RIPK3, and MLKL, and impeded the growth of in situ pancreatic tumors in mice [122]. Different pH values can lead to distinct types of cell death; alkali poisoning has been identified as a new strategy for the treatment of various tumors. Song et al. [123] demonstrated that JTC801 (combined with an opioid receptor with analgesic properties) limited the activation of NF-κB in a pH-dependent manner, induced alkaliptosis in PC cells, and slowed tumor growth in mice.

Under the action of various compounds, they can also affect the necroptosis pathway and play a role in cancer therapy. SK (a naturally occurring naphthoquinone derivative) directly targets RIPK3 in AsPC-1 cells and induces necroptosis. In a PANC-1 xenograft model established in nude mice, SK treatment significantly reduced the tumor size. When combined with gemcitabine, the tumor volume further decreased, and the antitumor effect of gemcitabine was enhanced [124]. Another novel pyridazinone compound, IMB5036, promoted the swelling of PC cells, increased the membrane permeability, activated necroptosis, and inhibited the growth of human PC cells. Among them, MLKL inhibitors can weaken their cytotoxic effect on cancer cells [125]. Adiponectin (APN) has antidiabetic, anti-inflammatory, and antiangiogenesis properties in various diseases. In PC, its inhibitor, AdipoRon, mainly triggered rapid mitochondrial Ca^2+^ overload and then produced superoxide, causing RIPK1/ERK-dependent necroptosis in cancer cells and inhibiting the growth of PC [126]. Diabetes has been reported to promote PC and lead to PC-related diabetes [127]. Therefore, it is necessary to further study the efficacy of AdipoRon in the treatment of PC.

In addition, it has been found that electrochemical therapy (ECT) could increase the cell-membrane pore diameter and drug uptake by using low-pulse voltage to overcome the voltage, poor permeability, and chemoresistance of PC; enhance the local chemotherapy effect; and have less damage to surrounding healthy tissues. The efficacy of ECT in the treatment of skin and subcutaneous melanoma has already been established [128], and it may also be applied to enhance chemotherapy in other malignant tumors. In pancreatic cells, electrochemical therapy combined with chemotherapeutic drugs increases the cytotoxicity of the drugs, causing more necrotic morphological changes compared with single drugs. These cell deaths can be rescued by the caspase inhibitors zVAD.fmk (zVAD) and Nec-1, which implies that necroptosis participates in cancer therapy [129].

Collectively, the above research indicates that activating the necroptotic pathway inhibits the progression of PC, which is inconsistent with the findings of Seifert and Ando. This may be attributed to the different inducing factors, doses, and time windows of necroptosis, and differences in experimental reagents, methods, and human factors for detecting the key proteins of necroptosis. In addition, these studies were conducted through in vitro experiments or animal models, and the clinical feasibility was not evaluated in clinical trials. Therefore, in future studies, additional data should be collected to accurately reflect the role of necroptosis in the treatment of PC and provide sufficient evidence for the development of further necroptosis-related treatments.

## 5. Conclusions and Future Perspectives

In recent years, immunotherapy based on immune checkpoint inhibitors has achieved success in the clinical treatment of cancer. However, only one-third of people respond to these drugs. Due to the immune resistance of most tumors to apoptosis, the search for alternative cell-death mechanisms has gradually become a new therapeutic strategy for cancer treatment [130]. For chemotherapeutic and immunotherapeutic drugs with superior efficacy in other cancers, PC has not yielded favorable clinical results due to drug resistance. Therefore, attenuating apoptotic resistance and developing nonapoptotic forms of cell death as an alternative cancer therapy is an attractive option. Necroptosis is a new nonregulated cell-death mode that combines apoptosis and necrosis; it not only plays a key role in viral infection, but also in the regulation of inflammation and cancer biology [23]. It plays complex roles in cancer progression, metastasis, patient prognosis, immune regulation, cancer-subtype determination, and cancer therapy. Furthermore, the activity of RIPK1 is mediated by several posttranslational mechanisms, such as ubiquitination and phosphorylation, which positively or negatively regulate cell death and survival in inflammatory diseases [131]. It has been previously reported that RIPK3-dependent mitochondrial phosphatase 5 regulates the activation of dendritic cells through dephosphorylation and is involved in antitumor immune responses [132]. Therefore, we posit that necroptosis has therapeutic and prognostic potential in pancreatic inflammation and tumors. The discovery of biomarkers to identify necroptosis, the thorough study of the molecular mechanism and physiopathological roles of necroptosis, with a particular emphasis on crosstalk between other cell-death mechanisms and immune-system interactions, can further develop antitumor therapeutic targets. The following table highlights the role of necroptosis in cancer therapy in recent years (Table 2).

Given the severe complications and high mortality of SAP, and the poor prognosis of PC due to its high invasiveness and drug resistance, new treatments are urgently needed. As our understanding of necroptosis deepens, its unique molecular mechanism in pancreatic diseases is expected to become a new target for treatment (Table 3: Therapeutic drugs in pancreatitis and cancer). Due to inconsistencies in PC research, more attention should be paid to the application of necroptosis in PC in future studies, such as exploring the relationship between necroptosis and PC stem cells, immune cells, chemoresistance, and immune infiltration; the association with other forms of cell death; the interaction between RIPK3 autophosphorylation, oligomerization, and the MLKL in necroptosis signal transduction; and the therapeutic effects of its regulator under the influence of electric and magnetic fields in PC. In addition, we can also focus on the role of posttranslational modifications of RIPK1, RIPK3, and MLKL in regulating their localization, activity, conformation, oligomerization, and protein interaction in signaling and interaction in signaling, excavate regulators to remove posttranslational modifications and regular necroptosis, and target to combat cell necroptosis in treatment. Therefore, we postulate that necroptosis may play a promising role in the treatment and prognosis of pancreatitis and malignancies, and particularly with regard to drug resistance and the immune system, which has valuable therapeutic value for the treatment of PC.

## Figures and Tables

**Figure 1 biomolecules-12-00828-f001:**
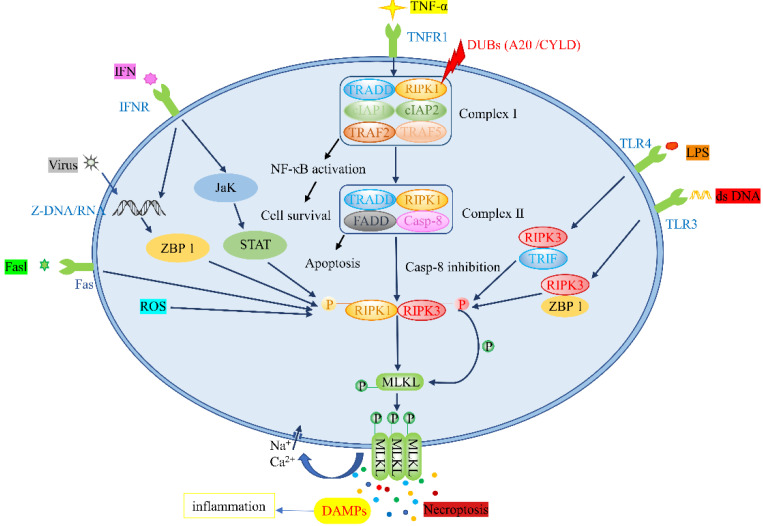
Pathways associated with necroptosis: CYLD: cylindromatosis.

**Table 1 biomolecules-12-00828-t001:** Comparison of Necroptosis with Other Forms of Cell Death.

Cell Death	Morphological Features	Biochemical Features	References
Necroptosis	Gradual translucency of cytoplasm and swelling of organelles, rupture of cell membranes	Phosphorylated MLKL translocates to the cell membrane and destroys the integrity of the cell membrane	[2,29]
Apoptosis	Plasma-membrane infiltration, mitochondrial-outer-membrane permeabilization, DNA fragmentation, nuclear disintegration	Cells break down into apoptotic bodies	[48]
Pyroptosis	Mitochondria remain intact, cells swell, and plasma membrane ruptures	Gasdermin family proteins, which form membrane pore upon proteolytic cleavage by caspases, granzymes, and microorganism-derived enzymes	[50]
Ferroptosis	Loss of plasma-membrane integrity, swelling of cytoplasm, organelles, chromatin condensation (mitochondria)	Iron accumulation, lipid peroxidation	[54]
Autophagy cell death	Cell contents are transported to lysosomes through double-membrane vesicles (autophagosomes) for degradation	Autophagosome formation, increased lysosomal activity	[55,56]
Necrosis	Cell structure disintegrates, mitochondria deform and swell, and plasma membrane ruptures	Activation of RIPK3, the molecular mechanisms of MLKL-dependent and MLKL-independent necrosis	[57,58,59]
Parthanatos	DNA fragmentation, chromatin condensation	PARP is overexpressed, and mitochondria-associated AIF is ectopic	[60]
Entotic cell death	Intercellular adhesion, lysosomal fusion, internalized cell death, and degradation	E-cadherin expression, RhoA-GTPase, and ROCK activation	[61]
NETotic cell death	Release of nuclear and mitochondrial DNA	ROS accumulation	[62]
Immunogenic cell death	Changes in cell-surface components and release of soluble medium	TAAs, DAMPs, release of proinflammatory cytokines, antigen-specific immune responses	[63,64]
Copper-dependent death	/	Apolipoprotein aggregation, iron–sulfur cluster loss, protein-toxicity stress	[45]
Anoikis	/	An inappropriate type of ECM	[46]

Notes: PARP: poly(ADP-ribose) collectase; AIF: apoptosis-inducing factor; TAAs: tumor-associated antigens.

**Table 2 biomolecules-12-00828-t002:** Expression of necroptosis factor in cancer and its impact on cancer prognosis.

Cancer Type	Necrosis-Factor Expression	Effect on Tumor	References
Breast cancer	RIPK1 expression is decreased, and ZBP1 expression is increased	Loss of ZBP1 reduces tumor lung metastasis	[133]
Cervical cancer	MLKL expression is increased	Associated with poor prognosis	[134]
Pancreatic cancer	RIPK1, RIPK3, and MLKL expressions are increased	Promotes metastasis	[119,121]
Colorectal cancer	RIPK3 expression is increased	Promotes tumor progression	[135]
Stomach cancer	MLKL expression is decreased	Related to a shorter OS	[136]
Lung cancer	RIPK1, RIPK3, and MLKL expression levels are decreased	Associated with worsening DFS	[137]
Glioblastoma	RIPK1, RIPK3, and MLKL expression levels are increased	Associated with shorter OS and DFS	[138]
Head and neck squamous cell cancer	MLKL expression level is increased	Associated with lymph node metastasis, tumor progression, and shorter OS	[139]
Acute myeloid leukemia	RIPK3 expression level is decreased	Promotes tumor progression	[140]

Notes: OS: overall survival; DFS: disease-free survival.

**Table 3 biomolecules-12-00828-t003:** Therapeutic drugs in pancreatitis and cancer.

Types of Pancreatic Diseases	Drug	Regulatory Factors	References
Pancreatitis	Nec-1	RIPK1	[95]
	TAK-242	TLR4	[102]
	KN93	RIPK3/p-MLKL	[103]
Pancreatic cancer	CCT137690	RIPK1/RIPK3/MLKL	[122]
	SK	RIPK3	[124]
	IMB5036	MLKL	[125]
	AdipoRon	RIPK1	[126]

Notes: p-MLKL: phosphorylated MLKL.

## Data Availability

Not applicable.

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
