# Peer review of "Understanding Necroptosis in Pancreatic Diseases"

_biomolecules, 2022, doi:10.3390/biom12060828_

Round 1
Reviewer 1 Report
The review by He, et al. summarizes the recent knowledge on necroptotic mechanism of programmed cell death and offers an insight into necroptosis role in variety of pancreatic pathologies, ranging from distinct forms of pancreatic tissue inflammation/pancreatitis to pancreatic cancer. More importantly – and appealing for the readers’ audience – the authors discuss the recently emerged details of necroptosis molecular biology and glean into potential therapeutic options targeting this cell death avenue in pancreatic pathologies.
The review is clearly written, well organized, brings up many interesting aspects of necroptotic cell death, strikes the right balance between basic and translationary aspects of necroptosis biology – all assuring its appeal for the readership of “Biomolecules” and warranting its publication.
Two recommendations to improve the review quality:
1. The text abounds with abbreviations, most of which are expanded in different sections, while some are given no expansion at all (e.g. DAMPs are mentioned in line 89, but no explanation is given for what this abbreviation stands for; expectedly this is “damage associated molecular patterns”). Authors should really consider producing a separate table of all abbreviations used in the manuscript to append it either in the text or set is as a separate section.
2. Table 1 is one of the pivotal focal planes of the review and is very helpful to compare and contrast distinct forms of cell death. It is almost comprehensive, clearly missing a somewhat controversial but reported on numerous occasions form known as ‘anoikis’. Anoikis form of cell death related to the interactions between the cell and surrounding extracellular matrix and has importance in such processes as loss of contact inhibition, survival of exfoliated cells during metastasizing, etc. – all fitting with the overall theme of He, et al. review emphasizing pathologic processes that impact that pancreatic tissue. I suggest including a small paragraph on anoikis in section 2.2, with corresponding listing in Table 1.
Minor stylistic suggestions:
1. Line 128: “Cell death contributes to cellular and tissue homeostasis…”
2. Lines 428-429: “…has gradually become a new therapeutic strategy for cancer treatment.”
3. Line 457: “…and its function in inflammation and carcinogenesis is expected to become…”
4. Lines 460-465: a very long sentence that looks incomplete and should be revised.
Author Response
Dear Reviewer,
Thank you for your review of the manuscript. Based on your valuable suggestions, we have revised the problems in the article one by one.
1.The text abounds with abbreviations, most of which are expanded in different sections, while some are given no expansion at all (e.g. DAMPs are mentioned in line 89, but no explanation is given for what this abbreviation stands for; expectedly this is “damage associated molecular patterns”). Authors should really consider producing a separate table of all abbreviations used in the manuscript to append it either in the text or set is as a separate section.
In response to your mention of abbreviations, we have checked the abbreviations present in the full text and added a summary table of abbreviations at the end of the manuscript text.
2. Table 1 is one of the pivotal focal planes of the review and is very helpful to compare and contrast distinct forms of cell death. It is almost comprehensive, clearly missing a somewhat controversial but reported on numerous occasions form known as ‘anoikis’. Anoikis form of cell death related to the interactions between the cell and surrounding extracellular matrix and has importance in such processes as loss of contact inhibition, survival of exfoliated cells during metastasizing, etc. – all fitting with the overall theme of He, et al. review emphasizing pathologic processes that impact that pancreatic tissue. I suggest including a small paragraph on anoikis in section 2.2, with corresponding listing in Table 1.
In response to your mention of anoikis, we have added this aspect to the manuscript by reading the relevant literature.
3: Minor stylistic suggestions
In response to your suggestion of minor stylistic suggestions, the corresponding part of the manuscript has been revised.
Please refer to the attached document for the revised manuscript.
Sincerely yours,
Ru He
Reviewer 2 Report
In this paper, the authors summarized the molecular mechanism of necroptosis and its influence on the development of pancreatitis and pancreatic cancer, and explored the effect of related molecular compounds in the treatment of diseases, in order to provide a foundation for the development of effective treatment methods for pancreatic diseases in the later stage. The manuscript is straightforward, well written, and concise and has clear results sithin the scope of a review article. It is a valuable contribution to the “biomolecules” journal. Some minor comments need to be addressed.
[1] “2.1.3. ZBP1-mediated pathway”, Page 3 of 26, Lines 116-119:
“Studies have also validated that IFN activates the RIPK1-RIPK3 necrosome complex via Jak/STAT signaling transduction to induce necroptosis when FADD is absent or not phosphorylated or when caspases are inactivated [40].”.
At that point should be incorporated an example. I would recommend to be reported that Jak/STAT 3 is a promising therapeutic target for ovarian cancer, as it is frequently overexpressed. Jak-stat 3 plays a role in tumour angiogenesis, survival, invasion, and chemoresistance. Inhibition of this pathway is shown to suppress tumour growth and progression.
Recommended reference: Ghose A, et al. Applications of Proteomics in Ovarian Cancer: Dawn of a New Era. Proteomes 2022, 10, 16.
[2] “2.2. Other forms of cell death”, Page 4 of 26, Lines 143-146:
“Furthermore, it is related to innate immune diseases and regulates the proliferation and migration of cancer regulated via non-coding RNA and other molecules, which has major implications for the prevention and treatment of cancer [49].”.
At that stage, the authors should highlight that identification of miRNAs and long non-coding RNAs can provide valuable information in diagnosis and prognosis and offers predictive value in several malignancies, such as melanoma. Their increase can be related to either the cancer or other conditions, such as inflammation. Therapeutic targeting of miRNAs can impact the natural history of melanoma by enhancing sensitivity to both standard therapies and immune checkpoint inhibitors.
Recommended reference: Revythis A, et al. Unraveling the Wide Spectrum of Melanoma Biomarkers. Diagnostics (Basel). 2021;11(8):1341.
[3] General comment:
A workflow diagram for the search would be of benefit for the readers.
Author Response
Dear Reviewer,
Thank you for your review of the manuscript. Based on your valuable suggestions, we have revised the problems in the article one by one.
[1] “2.1.3. ZBP1-mediated pathway” and [2] “2.2. Other forms of cell death”.
For your suggestion of these two sections add relevant literature. We found that adding these examples enriched the article by reading the relevant literature. Therefore, we have also added literature citations as you suggested.
[3] General comment: A workflow diagram for the search would be of benefit for the readers.
We can't quite understand what you mean by this suggestion. No corresponding changes have been made in the manuscript. I wonder if you can provide a template on your side for our reference.
Please refer to the attached document for the revised manuscript.
Sincerely yours,
Ru He
Reviewer 3 Report
In the review manuscript by He et al., the authors describe the molecular mechanism of necroptosis, and the contribution of necroptosis in pancreatitis and pancreatic cancer in a comprehensive manner. The manuscript is well-organized and clearly written. However, the terminology of some molecules should be confirmed. Therefore, I would like to recommend this manuscript for the publication in Biomolecules after the consideration of minor points.
MINOR POINTS
Both RIP1/RIPK1 and RIP3/RIPK3 are used throughout the manuscript. As RIP1 is identical to RIPK1 and RIP3 is identical to RIPK3, please use either RIP1/RIP3 or RIPK1/RIPK3, but not both.
In Figure 1, the authors illustrated ZBP1 and DAI as independent molecules, but these two proteins are identical. I recommend using ZBP1 rather than DAI because the former is more popular.
In Table 1, the authors described the biochemical features of Pyroptosis as “caspase-and granzyme-mediated activation of pepsin”. However, pepsin is a gastric digestive enzyme not related pyroptosis. This should be “gasdermin” family proteins, which form membrane pore upon proteolytic cleavage by caspases, granzymes, and microorganism-derived enzymes.
In Table 1, “Autophagy” should be “Autophagic cell death” because the primary function of autophagy is to maintain cellular homeostasis.
In Table 1, the authors described the biochemical features of necrosis as “activation of RIP3”. However, accidental necrosis does not necessarily require RIPK3 activation.
In the title of Table 2, “necrosis” should be “necroptosis”.
Table 3 is slightly confusing. For pancreatitis, regulatory factors are all direct targets of the drugs. In contrast, for pancreatic cancer, regulatory factors are downstream molecules and not direct targets of the drugs. For example, the direct targets of JTC801 are opioid receptors, and the authors of Ref.118 concluded that JTC801 lowered the expression of carbonic anhydrase 9 through NF-κB to induce “alkaliptosis”. Therefore, the effects of drugs against pancreatic cancer may be cancer-specific and complex, and the authors should reconsider the description of this table.
Lines 33-35, periods should be commas, and “V”,”M”, “L”, and “C” should be lower cases.
Line 194, “2+” should be superscript.
Lines 201-202, “p56” should be “p65”.
Line 434, necroptosis is a “regulated” cell death mode by RIPK1/RIPK3/MLKL.
Author Response
Dear Reviewer,
Thank you for your review of the manuscript. Based on your valuable suggestions, we have revised the problems in the article one by one.
1.Both RIP1/RIPK1 and RIP3/RIPK3 are used throughout the manuscript. As RIP1 is identical to RIPK1 and RIP3 is identical to RIPK3, please use either RIP1/RIP3 or RIPK1/RIPK3, but not both.
Based on your suggestion, we have replaced all the RIP1/RIP3 that appear in the manuscript with RIPK1/RIPK3.
- In Figure 1, the authors illustrated ZBP1 and DAI as independent molecules, but these two proteins are identical. I recommend using ZBP1 rather than DAI because the former is more popular.
Based on your suggestion, we have replaced the DAI in the manuscript using ZBP1.
- In Figure 1, the authors illustrated ZBP1 and DAI as independent molecules, but these two proteins are identical. I recommend using ZBP1 rather than DAI because the former is more popular.
4.In Table 1, the authors described the biochemical features of Pyroptosis as “caspase-and granzyme-mediated activation of pepsin”. However, pepsin is a gastric digestive enzyme not related pyroptosis. This should be “gasdermin” family proteins, which form membrane pore upon proteolytic cleavage by caspases, granzymes, and microorganism-derived enzymes.
5.In Table 1, “Autophagy” should be “Autophagic cell death” because the primary function of autophagy is to maintain cellular homeostasis.
6.In Table 1, the authors described the biochemical features of necrosis as “activation of RIP3”. However, accidental necrosis does not necessarily require RIPK3 activation.
7.In the title of Table 2, “necrosis” should be “necroptosis”.
For these issues you raised, we have revised them in the manuscript.
- Table 3 is slightly confusing. For pancreatitis, regulatory factors are all direct targets of the drugs. In contrast, for pancreatic cancer, regulatory factors are downstream molecules and not direct targets of the drugs. For example, the direct targets of JTC801 are opioid receptors, and the authors of Ref.118 concluded that JTC801 lowered the expression of carbonic anhydrase 9 throughNF-κB to induce “alkaliptosis”. Therefore, the effects of drugs against pancreatic cancer may be cancer-specific and complex, and the authors should reconsider the description of this table.
In response to your mention of confusing tables in this section, we have revised the tables in the manuscript by reading the contents of the manuscript and the original article.
- Lines 33-35, periods should be commas, and “V”,”M”, “L”, and “C” should be lower cases.
Line 194, “2+” should be superscript.
Lines 201-202, “p56” should be “p65”.
For these issues you raised, we have revised them in the manuscript.
- Line 434, necroptosis is a “regulated” cell death mode by RIPK1/RIPK3/MLKL.
In response to your question, we have read through the contents of the manuscript and did not find the expression in line 434. Therefore, no changes were made in the manuscript.
Please refer to the attached document for the revised manuscript.
Sincerely yours,
Ru He